# Intersections of informal work status, gender and tuberculosis diagnosis: Insights from a qualitative study from an Indian setting

**Sobin George**[1]*, **T. S. Syamala**[2], **Aditi Paranjpe**[1], **Mohamed Saalim**[1]

**1** Centre for the Study of Social Change and Development, Institute for Social and Economic Change, Bengaluru, India, **2** Population Research Centre, Institute for Social and Economic Change, Bengaluru, India

* sobing@gmail.com

## Abstract

### Background

There is evidence that more than one third of tuberculosis (TB) cases in India go undiagnosed each year and it is more pronounced among female patients. While there are studies available on the socioeconomic, cultural and gender-related dimensions of TB diagnosis delays among female patients in India, intersections of gender, informal work and diagnosis delays are not sufficiently studied. The present study aims to fill this gap by examining the TB diagnosis delay that are linked to the contingencies of working in informal arrangements for women from lower socio economic background.

### Methods

The study draws on 80 qualitative in-depth interviews conducted among female patients from lower socio-economic background, who were working or recently stopped working in informal arrangements and undergoing Directly Observed Therapy, Short-course (DOTS) treatment in Bengaluru (India) city and 60 willing significant others of the patients. The participants were identified through a scoping survey that covered 188 female patients from 18 DOTS centres in the city.

### Findings

Other than the already known reasons for the delay in TB diagnosis for women such as normalisation of symptoms, stigma and the gender-related discrimination leading to low prioritisation of women's illness, the present study identifies reasons related to work informality. These are normalisation of symptoms as workplace health problems; work related concerns that restricted formal help seeking; non TB specific narratives of symptoms, often incorrectly assumed to be work related health issues or comorbidities and thus confounding the early accurate diagnosis by the medical personnel and shifting between formal and informal systems of help-seeking. Further, the study found that mere knowledge of TB symptoms did not always translate to early diagnosis for patients from the lower socioeconomic groups working in informal arrangements due to the fear of losing work and wages owing to hospital visits.

**Data Availability Statement:** The study is based on primary data collected through qualitative in-depth interviews. Data is available with the authors in the form transcriptions. This could be shared on

request after obtaining the permission from the IEC. Supporting information of minimal data is uploaded.

**Funding:** The study was funded by the Indian Council of Medical Research (ICMR), New Delhi (https://main.icmr.nic.in/). SG and TSS obtained the funding (Project No. 7/2016/ICMR-ICSSR-SBR). The funders had no role in study design, data collection and analysis, decision to publish, or preparation of the manuscript.

**Competing interests:** The authors have declared that no competing interests exist.

## Conclusions

The workplace focus, especially the informal sector where a huge majority of India's workforce is employed, is notably absent in the TB elimination programme. The study indicates the need to adopt a comprehensive approach in the ongoing TB elimination programme in India in which family, living environment and workplace should be integral parts.

## 1 Introduction

The elimination of tuberculosis (TB) by 2030 is an important target set as part of the sustainable development goal 3, which is "ensuring healthy lives and promoting well being for all at all ages". The stated goal of India's National Tuberculosis Elimination Program (NTEP) is "to achieve a rapid decline in the burden of TB, morbidity and mortality while working towards elimination of TB [tuberculosis] in India by 2025". However, the burden of tuberculosis continues to be high in India. India accounts for nearly 28% of the global TB burden [1]. The TB notification rate has increased in India by 19 per cent from 2020 and 2021 and the TB notification in 2021 was nearly 1.9 million [2]. Majority of these cases are reported from the working age group and out of which 59% were in the age group of 15–44 and 89% were from the age group of 15–69 [2]. Further, the India TB Report 2022 shows that nearly 1/3rd of these cases are females. There are several challenges including poverty; undernourishment; comorbidities that weaken the immunity and hence making population susceptible to TB infection; emergence of drug resistant and extremely drug resistant strains; the socio economic and cultural factors related to delayed diagnosis and treatment; poor treatment adherence, prophylaxis and the non-resilience of the health systems to deliver TB programmes due to crises such as the COVID-19 pandemic, existing in India to realise the goal of TB elimination [3–6].

Available estimates showed that more than one third of TB cases go undiagnosed, untreated or unreported each year in India [7]. One of the major obstructions in the early detection and completion of TB is found to be its deep rootedness in the social determinants such as poverty, hunger, malnutrition, availability of clean drinking water and sanitation, social stigma and the gender constructions of social stigma [8]. Further, there are several reasons including disease related ones, social, cultural and economic reasons, identified for the delay in the diagnosis of TB among poor socioeconomic sections in India. Patient's perception of symptoms as normal or *not so serious* was one of the major socio-cultural reasons for the delay that these studies identified [9]. Studies also identified that lack of proper knowledge and awareness led to delayed diagnosis of TB of people from poor socio economic background [4, 5] and patients tended to take longer routes for diagnosis that involve consultations of informal providers, chemists and qualified medical practitioners, leading to diagnosis delay [10].

The gender-related reasons of TB diagnosis delays assume significance in India, especially when there is evidence for the underreporting of TB among women [11] and for the gender differences unfavourable to women in notification rate, clinical presentations and treatment outcomes [12]. Women faced more barriers including physical, financial and socio-cultural in reporting TB and longer delays than men for diagnosing TB in India [13]. The gender-related reasons for delay in the diagnosis of TB for women included lower intra-household spending for women's health, social stigma, fear of labelling and family responsibilities that led poor women in India not to prioritise their health and resort to home remedies and non-formal help seeking [14, 15]. Further, studies showed that female patients exhibited more anticipated stigma associated with TB than males, which resulted in the delay in diagnosis and treatment [16, 17].

There is sufficient literature available on the socioeconomic, cultural and gender-related dimensions of TB diagnosis delay among female patients in India,; however, the intersections of gender, informal work and TB diagnosis delays are not sufficiently studied. Also, while tuberculosis and its linkages with conditions work of men are well discussed, there is a dearth of focus of such research on women in India. The reasons for this low focus on the interconnectedness of TB and women's employment mainly include the non acknowledgement of women's labour, their lower participation in income earning occupations and the lower reported incidence of TB among women as compared to men along with a possible underreporting of TB among women. However, in urban and semi urban areas, the female labour market participation showed an increase due to the expansion of the lower level jobs in the informal economy, entry of more educated women in service sector jobs and the forced entry of women in informal labour market due to the decline in household income [18]. Further, as much as 91% of the paid jobs of women are in the informal sector without any social security entitlements, job security and most importantly, in hostile working conditions for health [18]. Most of the occupations of women in informal sectors in urban and semi urban areas are located in places such as households, schools, kindergartens, small factories and retail related service sector jobs where close interactions with others are frequent. The entry of more number of women in informal, low paying and less protected sectors, which have implications for infection, early diagnosis, treatment adherence and management of infectious diseases like TB is what makes the present study significant. The present paper hence aims to understand the linkages of gender and informal work status of women in the diagnosis of TB.

The paper includes the following. The first section based on a review of existing studies highlights the gap and the state the need to undertake the study. The second section discusses the methodology of the study in detail. The third section presents the main results, which are organised as three major themes on the delay of diagnosis. The fourth section offers a discussion based on the results and insights gathered from available secondary evidences and the concluding section while summarising the findings offers the measures to be taken to reduce avoidable delays in the early diagnosis of TB for female patients working in the informal arrangements.

## 2 Methods and materials

### 2.1 Participant recruitment and data collection

We employed a qualitative study among female working and recently stopped working TB patients in the city of Bengaluru, South India from January to December 2019. We undertook a scoping study before recruiting the participants for the qualitative study. To begin with we obtained the Directly Observed Therapy, Short-course (DOTS) directory with the contact details of key personnel from all four zones in the city. From this, we identified female TB patients who started treatment from January 2019 from selected DOTS centres. However, in some cases we were not able to find the required number of TB positive females. Thus wherever fell short of the target number, we migrated to the closest DOTS centre. We also checked wherever the next highest number of TB positive females was present and approached those DOTS centres for the interviews. We covered a total of 18 DOTS centres across four regions of the city and identified 188 female TB patients who underwent treatment from January 2019 to May 2019 for the scoping study. The criteria of selection were primarily their working status (as presently working or recently stopped working) and willingness to participate. From this group the study has recruited 80 participants for in-depth interviews, based on the criteria of data saturation. An interview guide was prepared for conducting in-depth interviews, which

included questions on participants' socio, economic, work and demographic background as well as the diagnosis and treatment seeking pathways.

Further, we interviewed significant others of 60 patients who participated in the in-depth interviews using an interview guide. We could not interview significant others of 20 patients since they did not share their disease information with anyone. A checklist was prepared for interviewing the significant others. The questions included their understanding of TB (symptoms, infectiousness, treatment and prophylaxis), decision making in health seeking in the family and their perceptions of the patients working with TB.

## 2.2 Data analysis

All interviews were recorded and the verbatim was transcribed and translated to English by bilingual experts. Sufficient care was taken to reproduce the information in participants' own words, phrases and expressions, which is a prerequisite of qualitative data analysis [19]. We developed seventeen codes, which are categorised under three code families. Axial coding was undertaken to draw connections between the different codes. Themes were then generated from these codes and code families (see Fig 1).

## 2.3 Ethical considerations

The study was approved by the Ethics Committee of the organisation where the authors are affiliated with (No. DPA/120/. . . ../2017). The consent of the participants to voluntarily take part in the study was obtained using a written consent form prepared in the local language of *Kannada*, which was signed by the participants in the presence of the investigators. The study did not recruit minor participants. The data was anonymised in order to keep the identity of the patients and significant others of the patients who participated in the study. The authors do not have access to the to information that could identify individual participants during or after data collection.

## 3 Results

The general background of the patient participants of the study is presented in Table 1. Out of 80 participants, 55 were in the age group of 15–34 and 19 were in the age group of 35–44. Most of them were educated with different levels of education ranging from primary to graduation and above. There were ten non-literates in the sample. Majority of them were either working in informal sector or as casual workers in formal sectors except for two patients out of which one was employed in public sector and other was an advocate. Out of the 80 participants, twenty six were employed in service related jobs, twenty five were working in factories, eighteen were in care/domestic work, six were professionals and five were self employed. Among those in the service related jobs, eleven participants worked as sales girls in retail shops; seven worked in offices in different jobs including receptionist, data entry operator and peon and remaining worked in hotels and restaurants (as helpers), hospitals (cleaners) and banks. Out of those who worked in factories, majority worked in garment factories followed by *Agarbathi (incense stick)*, silk, book binding and mirror manufacturing factories. Those who were self employed were mostly in street vending and running small shops (such as laundries, cloth pressing on wheels and haircut salons). There was one staff nurse, one advocate, two *Anganwadi* (kindergarten) teachers and one government employee in the category of professional/ regular employee. The sample contained 35 patients with pulmonary and 45 with extra pulmonary tuberculosis. There were three multidrug resistant (MDR) cases. Six patients in the sample had diabetes mellitus and one patient was infected with HIV. The HIV patient had MDR TB. There were 11 patients with recurred TB status in the sample.

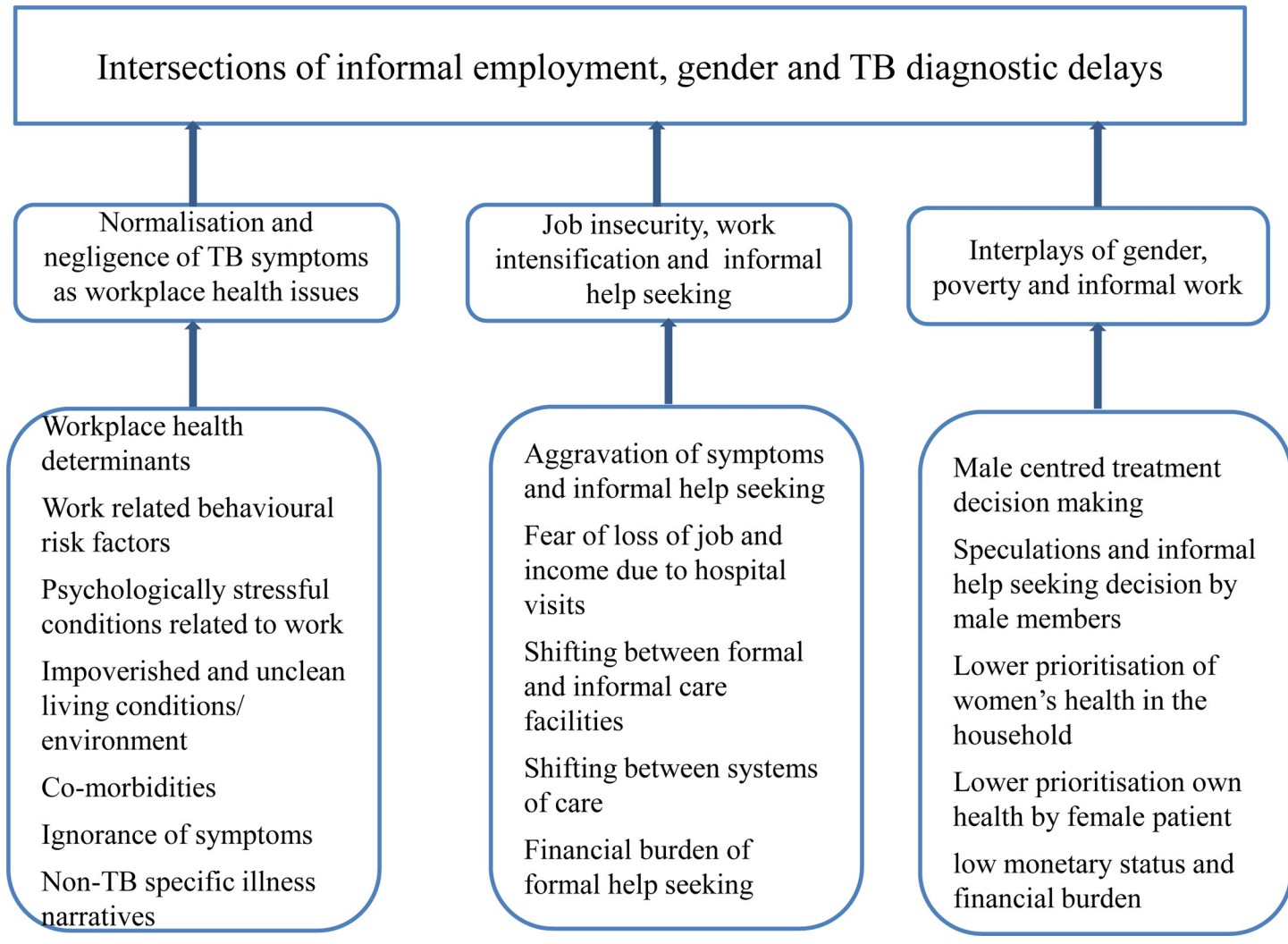

**Fig 1. Codes, code families and themes developed from qualitative data.**

### 3.1 Help seeking pathways and diagnosis delay of female TB patients

The help seeking pathways from the date of first symptoms to DOTS referrals of 80 participants who participated in the in-depth interview are presented in Fig 2. Most of the participants (N = 50) relied initially on informal sources of help that included chemists, home remedies and Registered Medical Practitioners (RMPs) as well as local *ayurvedic* clinics and shops. A few among them also relied on multiple informal health providers. Their diagnosis pathways showed a common trajectory of initially neglecting the symptoms for a brief period of time (ranging from one to three months), over the counter usage of medication for the symptoms of cough and fever and consultation of local RMPs/traditional practitioners in local private clinics. Final diagnosis was done at formal healthcare facilities, both in private and public hospitals. It was observed that TB was diagnosed mostly at public allopathic hospitals except for 16 participants who were diagnosed at private/charity hospitals. Further, all participants who sought help at public allopathic hospitals upon observing symptoms, were diagnosed for TB without requiring further referrals. A few participants who initially sought formal help at private hospitals shifted later to public hospitals where the diagnosis was done. All participants

**Table 1. Details of female working patient participants participated in in-depth interviews.**

|  | No | Respondent code |
|---|---|---|
| **Age** | | |
| 15–24 | 28 | P3, P8, P9, P14, P15, P23, P21, P31, P36, P42, P43, P44, P47, P50, P51, P52, P53, P54, P56, P60, P66, P68, P69, P73, P75, P76, P78, P80 |
| 25–34 | 27 | P1, P4, P7, P11, P12, P13, P17, P19, P22, P23, P24, P27, P28, P29, P30, P33, P34, P35, P49, P57, P61, P62, P63, P65, P70, P71, P72 |
| 35–44 | 19 | P2, P5, P6, P10, P16, P20, P25, P32, P37, P39, P40, P41, P45, P46, P4, P55, P58, P74, P79 |
| 45–54 | 3 | P38, P64, P67 |
| 55–64 | 3 | P26, P47, P77 |
| **Religion** | | |
| Christian | 2 | P19, P38 |
| Hindu | 66 | P2-P5, P7-P18, P20-P24, P26-, P30, P33-P35, P39-P59, P61-P68, P70-P72, P74-77, P79-P80 |
| Islam | 12 | P1, P6, P11, P25, P31, P32, P36, P37, P60, P69, P73, P78 |
| **Education** | | |
| Non-literate | 10 | P1, P7, P26, P33, P37, P38, P59, P74, P75, P77 |
| Primary | 9 | P13, P19, P21, P39, P40, P41, P45, P57, P78 |
| Secondary | 31 | P2, P3, P4, P5, P6, P9, P10, P11, P12, P16, P20, P23, P24, P28, P30, P32, P36, P44, P47, P49, P50, P51, P54, P55, P56, P60, P67, P68, P69, P73, P79 |
| Higher secondary | 13 | P14, P15, P17, P18, P29, P31, P34, P43, P46, P58, P61, P70, P76 |
| Graduation and above | 17 | P8, P22, P25, P27, P35, P42, P48, P52, P53, P62, P63, P64, P65, P66, P71, P72, P80 |
| **Nature of work** | | |
| Personal care/domestic work | 18 | P24, P20, P39, P4, P21, P41, P57, P45, P75, P1, P37, P38, P40, P28, P7, P67, P5, P23 |
| Self employed | 5 | P26, P77, P22, P80, P35 |
| Employed in factories | 25 | P64, P2, P15, P18, P30, P28, P33, P50, P51, P52, P54, P55, P56, P61, P66, P7, P69, P73, P78, P31, P32, P16, P14, P68, P47 |
| Service sector | 26 | P49, P12, P8, P9, P10, P29, P42, P43, P44, P48, P59, P62, P70, P72, P76, P79, P11, P60, P36, P19, P27, P3, P17, P53, P58, P34 |
| Professionals | 6 | P65, P46, P63, P71, P6, P25 |
| **Disease details** | | |
| Pulmonary | 35 | P1, P6, P7, P8, P10, P15, P18, P21, P22, P26, P28, P31, P32, P35, P40, P41, P43, P46, P49, P50, P51, P54, P56, P58, P59, P61, P63, P68, P69, P70, P72, P74, P76, P77, P79 |
| Extra-pulmonary | 45 | P2-P5, P9, P11-P14, P16-17, P19-P20, P23-P25, P27, P29-P30, P33, P34, P36-P39, P42, P44-P45, P47-48, P52-53, P57, P60, P62, P64-P67, P71, P73, P75, P78, P80 |
| MDR TB | 3 | P59, P61, P70 |
| TB with HIV | 1 | P70 |
| TB with diabetes | 6 | P6, P17, P38, P49, P64, P77 |
| MDR TB+HIV | 1 | P70 |
| Recurring cases | 11 | P3, P21, P28, P29, P43, P54, P57, P58, P70, P74, P76 |
| New cases | 69 | P1, P2-P20, P22-P27, P30-P42, P44-P53, P55-P56, P59-P69, P71-P73, P75, P76-P80 |

who were diagnosed at public and private hospitals were subsequently referred to the DOTS centres for treatment.

The duration between the first observance of symptoms and TB diagnosis across type of TB and socioeconomic characteristics of participants is presented in Table 2. Nearly 10% of pulmonary and extra pulmonary cases took more than three months for the diagnosis from the initial observation of symptoms. However, all MDR cases in the sample were diagnosed within a period of three months from the initial observation of symptoms. While most of the

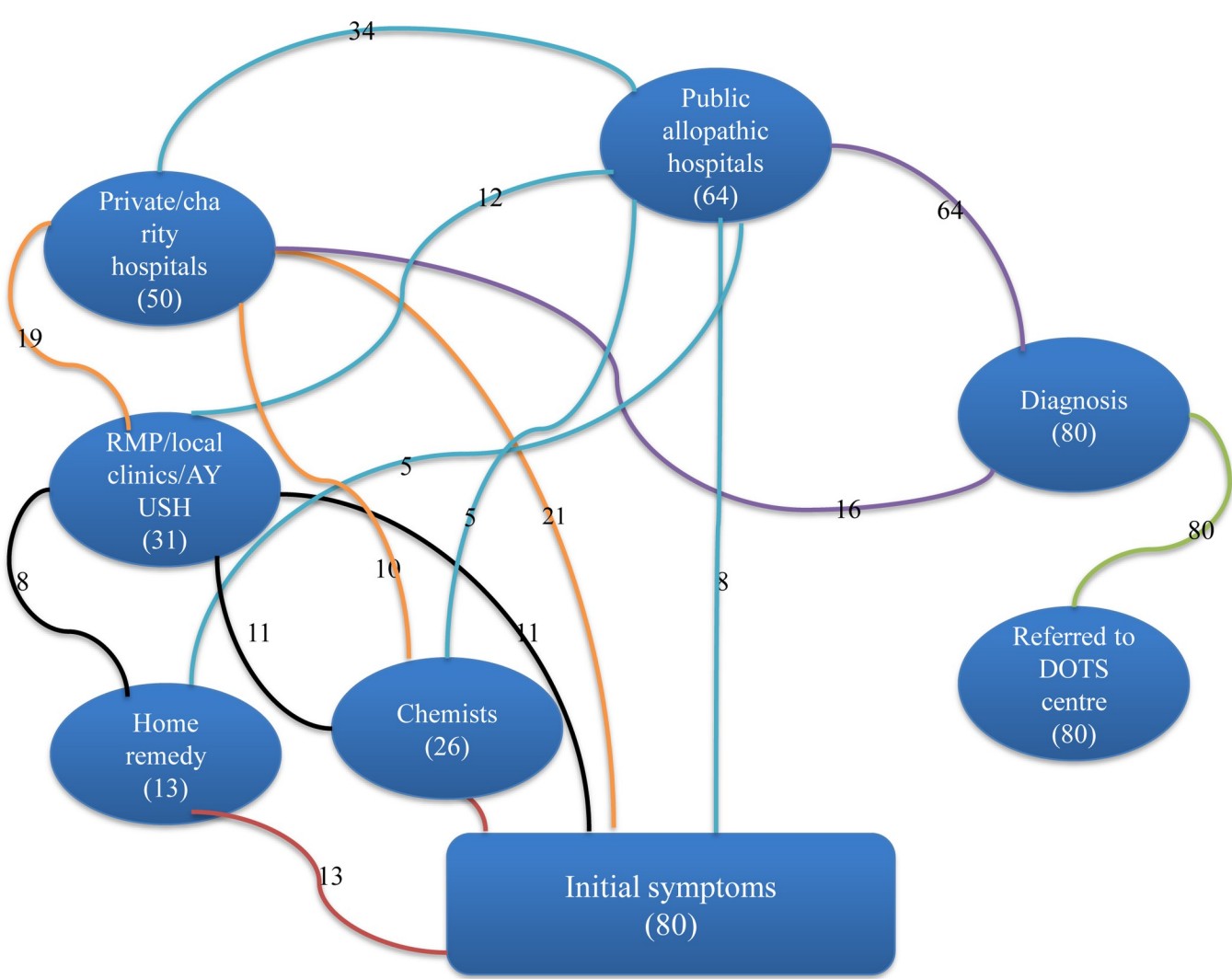

**Fig 2. TB diagnosis pathways of participants.**

participants were diagnosed within three months from the notice of symptoms, the duration of diagnosis varied across socioeconomic groups. For instance, across age groups, the diagnosis delay was found to be more in the group of 55–64, followed by in the group of 14 to 24. Similarly, delay was found more among participants from Christian community followed by Muslim, Scheduled Tribes, Scheduled caste, non-literate and widowed participants. It should also be highlighted that duration was longer for those who were employed at the time of interview than those who did not work.

We attempted to understand the interlinked reasons of delay in the diagnosis of TB for the working female patients in the larger canvas of informality of employment. Based on the narratives of participants we identified the major interlocking reasons for delay in the TB diagnosis of working females. These are presented in the following sections.

### 3.2 Normalisation of TB symptoms as workplace health issues

Normalisation of symptoms of TB, which led patients to neglect the symptoms in the initial period, came up noticeably in the narratives of most of the female patients who were/are

**Table 2. Duration till diagnosis from the notice of symptoms by background characteristics (N = 80).**

| Characteristics | Duration | | | | |
|---|---|---|---|---|---|
| | within a month | 1–2 months | 2–3 months | 3–4 months | More than 4 months |
| *Type of TB* | | | | | |
| Pulmonary | 17.9 | 33.2 | 28.7 | 9.9 | 10.4 |
| Extra Pulmonary | 9.9 | 18.9 | 33.4 | 28.0 | 9.8 |
| MDR TB | 20.0 | 37.0 | 32.0 | 11.0 | 0.0 |
| *Age* | | | | | |
| 14–24 | 16.8 | 31.0 | 26.8 | 9.2 | 16.1 |
| 25–34 | 18.6 | 34.4 | 29.7 | 10.2 | 7.1 |
| 35–44 | 17.8 | 32.9 | 28.4 | 9.8 | 11.1 |
| 45–54 | 17.4 | 32.2 | 27.8 | 9.6 | 13.0 |
| 55–64 | 16.4 | 30.3 | 26.2 | 9.0 | 18.2 |
| 65 plus | 20.0 | 37.0 | 32.0 | 11.0 | 0.0 |
| *Education* | | | | | |
| Non-literate | 16.8 | 27.2 | 26.8 | 13.1 | 16.1 |
| Primary | 20.0 | 46.9 | 21.6 | 11.5 | 0.0 |
| Secondary | 17.5 | 43.8 | 16.6 | 9.5 | 12.5 |
| Higher Secondary | 16.9 | 43.7 | 14.8 | 9.1 | 15.6 |
| Graduation and above | 19.4 | 55.3 | 14.1 | 8.4 | 2.8 |
| *Marital Status* | | | | | |
| Currently Married | 18.0 | 38.5 | 19.4 | 13.9 | 10.2 |
| Never Married | 18.3 | 39.8 | 19.8 | 13.6 | 8.5 |
| Divorced | 20.0 | 55.8 | 21.6 | 2.6 | 0.0 |
| Widowed | 16.4 | 34.4 | 17.7 | 13.3 | 18.2 |
| *Religion* | | | | | |
| Hindu | 18.5 | 43.8 | 19.9 | 10.2 | 7.7 |
| Islam | 17.6 | 41.7 | 19.0 | 9.7 | 12.0 |
| Christian | 12.5 | 29.6 | 13.5 | 6.9 | 37.5 |
| *Caste* | | | | | |
| SC | 17.9 | 42.5 | 19.4 | 9.9 | 10.3 |
| ST | 16.7 | 39.5 | 18.0 | 9.2 | 16.7 |
| OBC | 18.0 | 42.6 | 19.4 | 9.9 | 10.1 |
| General | 18.3 | 43.5 | 19.8 | 10.1 | 8.3 |
| *Status of current employment* | | | | | |
| Currently employed | 17.6 | 41.7 | 19.0 | 9.7 | 12.1 |
| Currently Unemployed | 18.1 | 43.0 | 19.6 | 10.0 | 9.3 |

working in informal arrangements. They related their illness to work and life and tended to normalise the TB symptoms such as cold, cough, fever, fatigue, loss of appetite, weight loss, body pain and swellings that they experienced. They attributed these symptoms to poor physical conditions at work such as dust and fumes; work intensification and associated issues such as physical and psychological stress and fatigue; work related behaviour risk factors such as skipping of meals and lack of time for rest and personal care and impoverished conditions of living such as lack of access to basic needs such as proper shelter, food and clean environment (Table 3).

A related aspect of normalisation and negligence of TB symptoms as part of the everyday experience of work and life of female patients was their illness narration, which were often non-TB specific. Other non-TB ailments which the patients had such as Asthma and Migraine

**Table 3. Patients' normalisation of TB symptoms related to work and living conditions.**

| Sub-themes | Qualitative illustrations |
|---|---|
| Trivialising and Neglecting symptoms | I used to get a fever everyday in the evening. I had the swelling near my ear for a year. I neglected them (P51). |
| Workplace health determinants | There is a lot of dust, fumes and the smell of chemicals [at the workplace]. Cough, headache and breathing problems are common (P2, P16, P30, P50, P56, P61, P78).<br>I must sit in one position throughout the day doing my work. I thought my problems [TB symptoms] were due to it (P56).<br>I have back pain due to work. I thought [TB] symptoms were due that (P7).<br>I had an accident at my workplace and had surgery. I thought it was the reason [of TB symptoms] (P29) |
| Work related behavioural risk factors | I skip meals often due to work pressure. If I have time then I eat else I do not eat. I lose weight sometimes, gain sometimes (P58).<br>I cannot take enough rest and sleep due to work (P3). I have no rest and recreation at all. I have no time for any personal care either. I have no energy and [am] always tired (P45).<br>It is hard to find time for personal care. I don't have time to take care of my health (P61). |
| Psychologically stressful conditions related to work | Often when I am depressed I don't care about myself at all (P58).<br>worried about achieving targets at work and loss of job (P2, P4, P5, P6, P7, P8, P9, P10, P11, P12P13, P15-P26, P34, P47, P59, P60, P75) |
| Impoverished and unclean living conditions/environment | I don't eat good food (P3, P28, P32, P33, P65).<br>I live in a slum. It's dusty and dirty (P28).<br>In and around my house there is a big open drainage due to which a lot of mosquitoes grow. Fever is common here, so I thought it is that (P40). |

for example, made both the patients and practitioners to attribute the symptoms to work-related or the other "non-TB illness related" concerns. This led to a neglect of the possibility of a TB infection. In such cases the patients underwent a complex pathway that included informal help seeking, outpatient medication for a longer period and non-TB specific diagnostic interventions, referrals and treatments. A 26 year old participant who worked as domestic help explained the delay in the diagnosis of extra pulmonary TB due to non-specific TB illness narratives and interventions by practitioners:

> I used to suffer from headache in the evenings. I also had high fever itching eyes, tiredness and weakness. I had migraine and told the doctor about it. I was treated for headache. Doctor advised for 6months treatment but these symptoms increased. So I left that hospital and kept on buying medicines from the medical shop for headache. I took *ayurvedic* treatment also in between. It helped me for a while but the headache came back. Then I was referred to . . . .[name of a private hospital]. The doctor there also said that it (pain) was migraine and gave medicines for 10days, which was extended to another 20 days. Pain got relieved initially but started again. Later I got a doubt that the extra growth in my neck might be causing the pain. So I showed that lump for the first time to the doctor he said immediately that it is Tuberculosis related growth (P24).

Further, it was found that instances of normalisations, trivialisations, negligence and lower prioritisation of TB symptoms found to have complicated the diagnosis and treatment for women working patients. The following case of a 40 year old participant who worked as a domestic help explains how her diagnosis of pulmonary TB got delayed and complicated due to normalisation and negligence of the actual TB symptoms by the patient:

I thought that it [fever] may be due to some allergy and took *shunthikashaya* [ginger decoction or reduction with other herbs]as home remedy. Then I went to private hospital. There they gave me injection and syrup. He [doctor] said take these [medicines] for one week. But I could not bear it at all by then. So after three days I went to a private hospital at Domlur [name of a place]. There they gave me an injection and then asked me to do blood check-up and an X-ray somewhere outside the hospital. It took nearly three months for me to diagnose TB and start the treatment (P41).

Similarly, a23 year old participant who worked in a retail shop of a shopping mall in the city elaborated on how multiple factors such as ignorance of the symptoms, own negligence, presenting symptoms that were less specific for TBled to the delayed diagnosis of her extra pulmonary TB:

At first I started coughing a lot. Then I developed severe vomiting and fever. Also I was getting pain in my right side of the body. Then I consulted a local doctor near my house. She went on giving me tablets for one whole year which was of no use to me. I got scanning done three-four times. And they also saw that the pancreas had a cyst. All these scans and treatments were done at various hospitals. I went to three places [names of hospitals]. No results! Finally I went to a public hospital where they did MRI and biopsy and diagnosed TB. (P42)

The presence of co-morbidities which have overlapping symptoms of TB was another reason for normalisation of TB symptoms. For instance, some of the patients also suffered from diseases like typhoid, migraine and goitre when they started showing the symptoms of TB. Since most of the symptoms were similar, both the patients and practitioners did not suspect the possibility of TB infection. Following narratives illustrate how patients' co-morbidities mimic the symptoms of TB and delay the diagnosis.

First I had fever. That was because I had typhoid. That time I consulted a private clinic nearby. They gave me treatment for typhoid. It went on for three months. But my cough did not subside. So they sent me to [name] Hospital. It was confirmed there that I had TB (P52)

I had very serious thyroid related issues in the past. I have had two surgeries for the same. I always used to experience pain at the back of my head. It was so bad that I could never even put my head on the pillow comfortably. I started feeling mentally depressed as well. Then my husband said that maybe this is connected with my thyroid problem. Hence we decided not to get a check up done. Around the same time I started getting some sort of bubbles under my neck. When we checked for my thyroid it was found that thyroid is normal. But we got several tests done since the symptoms continued and TB was diagnosed finally (P58).

## 3.3 Job insecurity, work intensification and informal help seeking

While the initial responses to TB symptoms were negligence, trivialisation and normalisation, patients began help seeking with the persistence or aggravation of symptoms (Table 4). The initial help seeking started with self treatment including home remedy, over the counter medication and then at a later stage the consulting with RMPs and/or *Ayurvedic* shops in the locality. This was due to the fear of losing employment and income owing to hospital visits. In

**Table 4. Factors led to non-formal help seeking.**

| Sub-themes | Qualitative illustrations |
|---|---|
| Aggravation of symptoms and informal help seeking | Initially I had cough for 3months I thought it is normal cough (P3). I had headache, sometimes fever, tiredness, I took over the counter medicine headache reduced not up to my expectations (P19) |
| Fear of loss of job and income due to hospital visits | I had a cough, fever and I went to a clinic. My husband and I were given fever medicine, then went to another clinic for the treatment for fever again for a week. Then I went to . . . . . .[name] hospital there was also nothing effective after that another clinic. I went to ..[name] Hospital and I was treated. It took two months to detect Tuberculosis (P35). |
| Aware of TB symptoms, but unable to seek formal help | I work in garments [factory]. There they never give me leave. Hence also I delayed going for a check-up due to the fear of loss of job and wages. Although I doubted TB, I did not have an option (P55). |

certain cases, in spite of patients being aware and attributing their symptoms to TB, they were yet unable to seek help at formal health facilities. It emerged from their narratives that this avoidance was because they were aware that seeking help could possibly lead to loss of working days. They, in most of the cases, were the sole earners of the family and the loss of working days and wages would adversely affect meeting the basic needs of the family. Hence they were left with the option of using non-formal care options such as local doctors(RMP), local practitioners and healers (sometimes even without formal qualification), or then medical stores.

## 3.4 Interplays of gender, poverty and informal work in formal help seeking

There were instances when erroneous judgments and speculations of family members, especially the dominant male member of the family who often takes treatment seeking decisions for others, led to prolonged diagnosis pathways for the female patient (Table 5). For instance, often incorrect beliefs of and speculations by the decision-maker took the patient to faith healers, self medication, consultations at local clinics and non-TB specific diagnostic and treatment interventions.

Interviews with significant others highlighted the overriding concerns of the family about the possible financial burden due to seeking treatment in private hospitals which led them to seek non-formal help for the initial symptoms of TB of the female patients in the family. In several such cases illness of female members were prioritised lower than that of the head of the household, male members and children in the family. Interviews with significant others of the patients further highlighted how the symptoms of the female patient were neglected and assumed to be "mere cough and fever". Due to the general difficulties caused by the low monetary status and financial burden experienced by these households, the health condition of the females often received very little or no attention. The meeting of day to day expenditures of the household, including food and education of children and managing the diseases of other family members were considered more important than these "normal" illnesses of the female member.

## 4 Discussion

The present study illustrates the linkages between the informal employment status of the women, the gender-related discrimination both overt and covert, and the delays in the diagnosis of TB. These linkages were seen to lead to long complex diagnosis pathways and thus delaying the diagnosis of pulmonary TB for two to three months and extra pulmonary TB for a period of more than four months. Their diagnosis pathways showed a trajectory starting with initial neglect of the symptoms for a brief period of time, relying on over the counter

**Table 5. Gender intersections of informal help seeking.**

| Sub-themes | Qualitative illustrations |
|---|---|
| Wrong convictions of decision making male member in the household | We were not aware that she had TB. She had a lot of pain in her leg. We assumed she had slipped somewhere. You know, we went to temples thinking something is wrong with the leg. We went to the oil shop [*ayurvedic* shops] as well. Hospitals also told, this could be because of gastritis and advised not to eat certain pulses. After that we went to . . ..[name] hospital. (SO4). |
| Lower prioritisation of diseases of women in resource-poor households | We have a lot of expenditure in the family. We are getting the children educated, we both aren't educated. We do not have our own house in Bangalore. We have to pay 4000–5,000 rent. We can't go to the hospital for every common disease (SO19, husband of the patient). My husband has a heart problem and we already spent a lot of money for that. What will we treat? His problem or her cough and fever? (SO31, mother of the patient) |

medication, seeking help with local RMPs, and then seeking help at private hospitals where often tests for unrelated health conditions were conducted, public or private hospitals where TB was diagnosed and then public hospitals/DOTS centres where treatment was finally begun. Further, it was observed that referrals to or direct help seeking at public allopathic hospitals led to faster diagnosis whereas the formal (except for public hospitals) and informal help seeking involving multiple sources/facilities tended to delay the diagnosis. Most importantly, participants, who were working, reported longer delay than those who stopped working, indicating the intersections of informal working status and TB diagnosis delays for female patients.

The reasons for TB diagnosis delays such as normalization of TB symptoms, informal help seeking, gender-related reasons such as lower prioritization of health of women within the households and lack of knowledge [3–5, 13–15, 20–22]were true for the participants of the present study as well. Besides these, the present study found that women working in informal arrangements tended to normalize TB symptoms as workplace health problems and delayed formal help seeking. Most of the patients who participated in the study were working/worked in informal arrangements. They worked as domestic helps, domestic care workers, housekeeping staff in schools, colleges, hospitals and private offices, sanitary workers, cleaners and helpers in restaurants, workers in garment, *Agarbathi*, book binding and glass factories, sales girls in retail shops, contract workers in Information Technology Enabled Sectors (ITES) (like telecalling and back end services), workers in beauty parlours, nursing staff and *Anganwadi* teachers. The adverse conditions of work at informal settings including absence of social security, job insecurity, absence of leave, work intensification, absence of collective bargaining and harassment at workplace are well brought out in the Indian context [23]. Except for some of the work environments including beauty parlours, *Anganwadis* and households, most of the workplaces were physically dangerous and non-conducive environments with poor ventilation, without facilities for rest and sanitation and with continuous exposure to dust and fumes. Hence, in most of the cases, meanings that patients attributed to the symptoms of TB were related to poor physical conditions at work, work intensification and associated issues such as occupational risks and work related behaviour risk factors, psychological stressful conditions related to work and impoverished conditions of living (See Fig 3).

Further, the study found that TB diagnosis for female patients who worked in informal arrangements were delayed even at formal health facilities since workplace health problems and occupational risks dominated the illness narratives of working female patients, without directly referring to the symptoms of TB. This led the medical practitioners to conduct non-TB specific diagnostic interventions in the initial period for diseases including migraine,

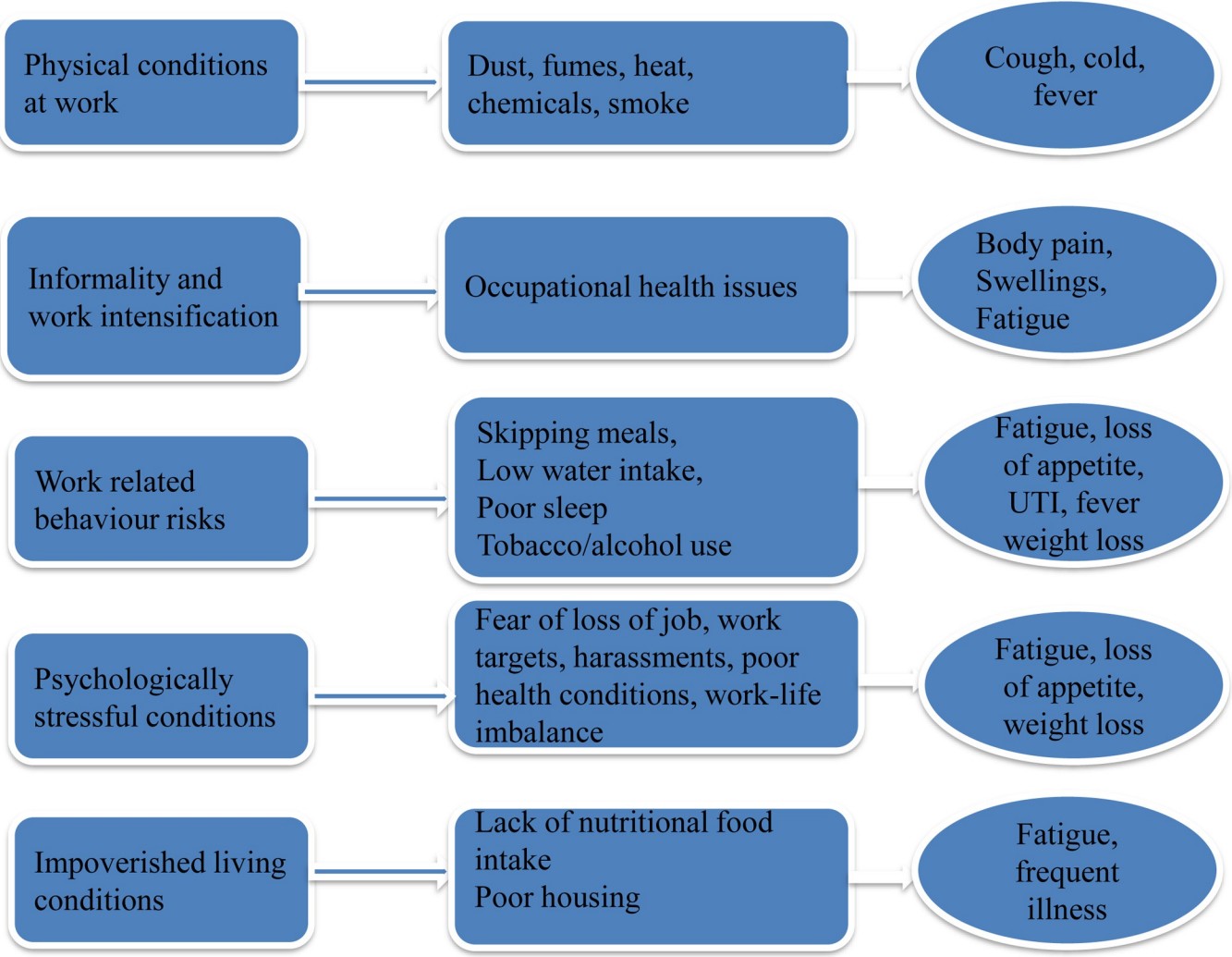

**Fig 3. Normalised meanings of TB symptoms associated with informal work.**

musculoskeletal disorders, asthma, cancer, heart disease and epilepsy, which not only increased the diagnosis delay but also the cost of treatment. In some cases, comorbidities of patients that have overlapping symptoms of TB misled the practitioners, resulting in TB diagnosis delay.

The study also found that although the knowledge of the disease was a necessary condition to seek treatment on time, it was not always sufficient for working women patients as in many cases knowledge of symptoms of TB did not translate to formal help seeking. Tuberculosis was not an unfamiliar disease for most of the participants. A few participants had prior knowledge/information on TB through others' experience of TB in the family or through their own previous episodes of infection. Moreover, TB was not an unusual disease in their neighbourhood (which in most of the cases was an urban slum). While the knowledge of TB, especially its symptoms, helped several patient participants to be alert and seek timely help in formal health facilities, a few of them, despite their prior knowledge of the disease symptoms, opted to seek help from informal sources available in the vicinity. It was found that the fear of loss of job and wages due to hospital visits were the major reasons that delayed the help seeking for these

patients. They were either piece-rated or daily wage workers and were not entitled to any social security measures or covered under statutory labour welfare provisions. They were the sole or principal earners in the family. Hence they availed the most feasible option of informal help seeking so that they would not lose days of work and wages. We note that mere awareness generation without addressing the real structural problem of poverty and work informality may not help patients to reduce the delay in diagnosis and complete treatment successfully.

The household decision making on help seeking, which were mostly done by the male/head of the family, was another important theme emerged in the study. Gender-related reasons were found to have interplayed with other factors leading to late diagnosis too, when the female patients' families tended to prioritise the household needs within their limited income. Narratives of a few family members reflected their overriding concerns about the possible financial burden due to treatment in private hospitals and associated costs (loss of wages of patient and accompanying person, travel, food) which led them to consult RMPs and local *ayurdevic* doctors for the initial symptoms of TB of the female patients in the family. In several such cases, illness of female TB patients were much lower in the order of priority than the needs of the head of the household, male members and children.

## 5 Conclusions

The study illustrated that working in informal arrangements contributed to the delay in the diagnosis of TB for female patients. The informal working arrangements that are characterised by job insecurity, absence of leave with pay, lack of social and health security and moreover physically dangerous and psychologically stressful conditions of work contribute to the delay in TB diagnosis of working women along with other reasons in a number of ways. While the already known reasons such as normalization of TB symptoms, informal help seeking, gender-related reasons such as lower prioritization of health of women within the households and lack of knowledge were true for those who delayed the initial formal help seeking, the study found that women working in informal arrangements normalized TB symptoms as workplace-related health problems. The study also found that although awareness of the disease can help patients to seek help in formal healthcare facilities, it may not be a sufficient condition for all patients due to concerns related to work and wages. Further, the study showed that the reliance upon multiple informal treatment sources, non TB-specific interventions and long procedures at formal health facilities can delay TB diagnosis. Other factors that delay the TB diagnosis of female patients working in informal arrangements are non-specific TB narratives, shifting of help-seeking between informal and formal health facilities, and the fact that TB mimics symptoms of other diseases.

The study indicates that there is a need to adopt a comprehensive approach in the ongoing TB elimination programme in India in which family, living environment and workplace should be integral parts. It emerged from the study that the workplace focus, especially the informal sector where a huge majority of India's workforce is employed, is notably missing in the TB elimination programme. One important step that could be taken is the dissemination of proper information about TB at the workplace. It is also important to address the issue of the fear of loss of employment and wages of patients working in the informal arrangements that delayed the formal help seeking and early diagnosis. The study suggests that the TB elimination programme could consider steps such as linking the informal sector workers to the existing Employee State Insurance (ESI) system, which can support the patents during the days without work. Also, a special wage allowance could be made part of the TB elimination programme, which can pay the wages of patients for their absence from work during the period of diagnosis and treatment.

## 6 Limitations and future areas of work

The paper has not analysed the data on the reasons of TB diagnostic delay across socio-demographic, economic and cultural background, which is a future area of research. Similarly, the paper has not presented the perspective of employers of the participants and DOTS service providers on the diagnostic delays. It is important to gather their views in order to further understand the implications of stigmatisation of employers and the systemic issues related to health services on TB diagnostic delay.

## Supporting information

**S1 Table. Baseline data.**
(DOCX)

**S2 Table. Socio-demographic data of participants.**
(XLSX)

## Acknowledgments

The authors thank Gautham Sathyaprem and Kusuma C R for their assistance in conducting the fieldwork.

## Author Contributions

**Conceptualization:** Sobin George.

**Data curation:** Mohamed Saalim.

**Formal analysis:** Sobin George, Mohamed Saalim.

**Funding acquisition:** Sobin George, T. S. Syamala.

**Investigation:** Aditi Paranjpe, Mohamed Saalim.

**Methodology:** Sobin George, T. S. Syamala.

**Project administration:** Sobin George, Aditi Paranjpe.

**Resources:** Sobin George.

**Software:** T. S. Syamala, Mohamed Saalim.

**Supervision:** Sobin George.

**Visualization:** Sobin George.

**Writing – original draft:** Sobin George.

**Writing – review & editing:** T. S. Syamala, Aditi Paranjpe.

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
