## [Decision Letter · Decision Letter 0]

9 Jun 2023

PONE-D-23-13272Intersections of informal work status, gender and Tuberculosis diagnosis: Insights from a qualitative study from an Indian settingPLOS ONE

Dear Dr. George,

Thank you for submitting your manuscript to PLOS ONE. After careful consideration, we feel that it has merit but does not fully meet PLOS ONE’s publication criteria as it currently stands. Therefore, we invite you to submit a revised version of the manuscript that addresses the points raised during the review process.

We look forward to receiving your revised manuscript.

Kind regards,

Praveen Kumar Balachandran

Academic Editor

PLOS ONE

Journal Requirements:

Additional Editor Comments:

Dear Authors,

The manuscript needs a revision as per the constructive comments given by the reviewers before getting accepted. Few changes need to be done for betterment.

Reviewers' comments:

Reviewer's Responses to Questions

**Comments to the Author**

1. Is the manuscript technically sound, and do the data support the conclusions?

Reviewer #1: Yes

Reviewer #2: No

2. Has the statistical analysis been performed appropriately and rigorously? 

Reviewer #1: Yes

Reviewer #2: No

3. Have the authors made all data underlying the findings in their manuscript fully available?

Reviewer #1: Yes

Reviewer #2: Yes

4. Is the manuscript presented in an intelligible fashion and written in standard English?

Reviewer #1: Yes

Reviewer #2: No

5. Review Comments to the Author

Reviewer #1: The general idea of the paper seems to be good. However, the paper organization and presentation are appreciable.

In my opinion, this article's findings could use some more elaboration.

Female patients in India, gender discrimination in the workplace, unpaid caregiving, and time lags in diagnosis are all topics that could benefit from further research.

I suggest the authors to provide the detailed study about data analysis and what are the metrics have taken in to the account while doing the survey of analysis.

All sampled MDR cases should have been diagnosed, and the minimum time required to determine the study's outcome—three months from the onset of symptoms—should be met.

In this study, authors may suggest transitioning from informal to formal arrangements based on their findings regarding why people ignore symptoms in the introductory period.

Intention and route map of figure 1 and 2 are good.

The paper's key contribution has to be stressed and recognised. It would be excellent if the disadvantages and gaps in the literature, and especially how the suggested strategy seeks to remedy these gaps, were made obvious.

Reviewer #2: The authors have done a good job on tuberculosis diagnosis, but this paper still needs improvement.

1. The abstract should always mention the rate of efficacy/efficiency percentage of the proposed method for the reader’s quick overview.

2. The abstract should at least have a line or two about the need for this work. This abstract has an intro and it straightaway deals with the proposed work.

3. Apart from giving the literature in paragraphs, better to add a table which precisely shows the proposed method, pros and cons.

4. At the end of the introduction section, there should be ‘Paper Structure / Paper Organization’.

5. The novelty of this paper is not clear. The difference between the present work and previous works should be highlighted. Add more of the issues and what is the significance of this research.

6. Before the paper's contributions, the author could precisely include the need to develop the proposed method.

7. The authors proposed architecture could be more precise, and a flow/sequence diagram with every step involved shall give it.

8. Analytical data representation should be more in-depth and comprehensive.

9. Among 22 references, hardly one is from 2023. This shows that the paper hasn’t considered more contemporary related works in the survey. Other than tuberculosis some common disease prediction should be mentioned in the literature. I suggest a few more papers to cite and refer.

https://doi.org/10.1038/s41598-023-35922-x

https://dx.doi.org/10.2174/1574893618666230206121127

DOI: 10.1504/IJDATS.2022.10053183

https://doi.org/10.1111/exsy.13298

10. A separate section for Limitations and future work in detail would give further ideas for the readers who wish to enhance your work.

6. PLOS authors have the option to publish the peer review history of their article (what does this mean?). If published, this will include your full peer review and any attached files.

Reviewer #1: **Yes: **Dr.R.Karthikeyan

Reviewer #2: No

---

## [Author Response · Author response to Decision Letter 0]

6 Jul 2023

Please find below the responses and compliances to the comments of reviewers

The abstract should at least have a line or two about the need for this work. This abstract has an intro and it straightaway deals with the proposed work. 

Response: Added two sentences on the need of the work

The abstract should always mention the rate of efficacy/efficiency percentage of the proposed method for the reader’s quick overview.

Response: The present paper is based on qualitative methods and the rate of efficacy/efficiency percentage of the proposed method cannot be computed

At the end of the introduction section, there should be ‘Paper Structure / Paper Organization’. 

Response: Included a paragraph on paper structure/organisation

The novelty of this paper is not clear. The difference between the present work and previous works should be highlighted. Add more of the issues and what is the significance of this research. 

Response: Added a paragraph to emphasize the novelty of the paper

Before the paper's contributions, the author could precisely include the need to develop the proposed method. 

Response: Included

The authors proposed architecture could be more precise, and a flow/sequence diagram with every step involved shall give it. 

Response: Included another diagram that shows the architecture related to development of themes from codes and code families (figure 1)

Analytical data representation should be more in-depth and comprehensive. 

Response: Added more data and analysis in the 'results' section 

Among 22 references, hardly one is from 2023. This shows that the paper hasn’t considered more contemporary related works in the survey. Other than tuberculosis some common disease prediction should be mentioned in the literature 

Response: Cited 3 more relevant articles which are published in 2023

A separate section for Limitations and future work in detail would give further ideas for the readers who wish to enhance your work. 

Response: Included

---

## [Editor Report · Decision Letter 1]

12 Jul 2023

Intersections of informal work status, gender and Tuberculosis diagnosis: Insights from a qualitative study from an Indian setting

PONE-D-23-13272R1

Dear Dr. George,

We’re pleased to inform you that your manuscript has been judged scientifically suitable for publication and will be formally accepted for publication once it meets all outstanding technical requirements.

Kind regards,

Praveen Kumar Balachandran

Academic Editor

PLOS ONE

Additional Editor Comments (optional):

1. The authors revised the manuscripts according to the comments.

2. All the suggested corrections given by the reviewers were incorporated and addressed well.

3. I recommend this manuscript may be accepted for publication in the present form.

Thank you for the contribution.
---

## [Editor Report · Acceptance letter]

18 Jul 2023

PONE-D-23-13272R1 

Intersections of informal work status, gender and Tuberculosis diagnosis: Insights from a qualitative study from an Indian setting 

Dear Dr. George:

I'm pleased to inform you that your manuscript has been deemed suitable for publication in PLOS ONE. Congratulations! Your manuscript is now with our production department. 

Kind regards, 

on behalf of

Dr. Praveen Kumar Balachandran 

Academic Editor

PLOS ONE